# Breastfeeding and Sociodemographic Determinants: Evidence from the “MAMI-MED” Cohort

**DOI:** 10.3390/nu17162702

**Published:** 2025-08-20

**Authors:** Giuliana Favara, Andrea Maugeri, Martina Barchitta, Roberta Magnano San Lio, Maria Clara La Rosa, Claudia La Mastra, Erminia Di Liberto, Fabiola Galvani, Elisa Pappalardo, Carla Ettore, Giuseppe Ettore, Antonella Agodi

**Affiliations:** 1Department of Medical and Surgical Sciences and Advanced Technologies “GF Ingrassia”, University of Catania, 95123 Catania, Italy; giuliana.favara@unict.it (G.F.); andrea.maugeri@unict.it (A.M.); martina.barchitta@unict.it (M.B.); robertamagnanosanlio@unict.it (R.M.S.L.); mariaclara.larosa@unict.it (M.C.L.R.); claudia.lamastra@unict.it (C.L.M.); erminia.diliberto@unict.it (E.D.L.); 2Department of Obstetrics and Gynaecology, Azienda di Rilievo Nazionale e di Alta Specializzazione (ARNAS) Garibaldi Nesima, 95124 Catania, Italy; fabiolagalvani38@gmail.com (F.G.); elypappalardo@yahoo.it (E.P.); carla.ettore@hotmail.it (C.E.); giuseppe.ettore@gmail.com (G.E.)

**Keywords:** breastfeeding, pregnancy, maternal nutrition, social determinants

## Abstract

**Background/Objectives**: Breastfeeding is key to maternal and child health, but adherence to WHO recommendations varies worldwide and is influenced by several maternal and paternal factors. In this study, we aim to describe the prevalence of breastfeeding practices and adherence to WHO guidelines among women, and to explore the maternal and paternal characteristics associated with these practices. **Methods**: Data were obtained from the “MAMI-MED” cohort, which included women enrolled during the first trimester of pregnancy at ARNAS Garibaldi Nesima in Catania (Italy). Breastfeeding practices and parental characteristics were assessed through interviews conducted at 12- and 24-month follow-ups. **Results**: The analyses involved 1312 women enrolled between December 2020 and October 2023. Mothers who breastfed, particularly those who exclusively breastfed for the first six months, showed a more favorable socioeconomic profile. Women with a medium (OR = 1.781; 95% CI: 1.258–2.521; *p* = 0.001) and high level of education (OR = 3.892; 95% CI: 2.255–6.718; *p* < 0.001), as well as those who had a spontaneous delivery (OR = 1.461; 95% CI: 1.090–1.958; *p* = 0.011), were more likely to breastfeed. Similarly, adherence to WHO recommendations was higher among women with a medium (OR = 2.144; 95% CI: 1.339–3.433; *p* = 0.002) and high education levels (OR = 2.611; 95% CI: 1.489–4.580; *p* < 0.001), non-smokers (OR = 2.256; 95% CI: 1.158–4.395; *p* = 0.017), and those with adequate gestational weight gain (OR = 1.506; 95% CI: 1.035–2.189; *p* = 0.032). **Conclusions**: Sociodemographic and behavioral factors, particularly maternal education, smoking status, mode of delivery, and gestational weight gain, significantly influence breastfeeding practices and adherence to WHO recommendations. These findings highlight the importance of targeted interventions to support breastfeeding, especially among women with less favorable socioeconomic profiles.

## 1. Introduction

Breastfeeding is considered an effective preventive strategy for improving child survival, health, and optimal growth, as well as the well-being of lactating mothers [1]. The World Health Organization (WHO) recommends to begin breastfeeding infants within the first hour after birth and to exclusively breastfeed for the first six months of life, followed by continued breastfeeding alongside complementary foods up to two years of age or beyond [2]. Breastfeeding is also recognized as a critical component in the early prevention of childhood and adolescent obesity [3]. Effective prevention requires a multifactorial and multidisciplinary approach, promoting healthy behaviors from the earliest stages of life—such as breastfeeding and adequate nutrition—alongside interventions targeting biological, environmental, and social determinants [4,5,6]. Despite its recognized benefits throughout the life course, breastfeeding practices and adherence to global recommendations vary widely across regions [2].

In Italy, breastfeeding promotion is also supported by national legislation (Legislative Decree No. 151/2001), which grants employed mothers five months of fully paid maternity leave and, during the first year of the child’s life, two paid daily breastfeeding breaks (one for part-time workers), which can be combined. Additional protections include restrictions on night work and the possibility of reduced working hours under certain circumstances, aiming to facilitate breastfeeding continuation after returning to work. Data from the Italian surveillance system showed a high degree of regional variability in exclusive breastfeeding rates, with significantly lower proportions in the south compared to the north. At 2–3 months of age, exclusive breastfeeding rates ranged from 54.0% in northern regions to 36.4% in the south, and by 4–5 months, they ranged between 35.8% and 19.6% [7]. A wide range of maternal factors may influence women’s breastfeeding behavior, knowledge, and awareness about existing recommendations on breastfeeding practices [5]. Among these, maternal sociodemographic cultural and personal characteristics (i.e., age, education, and employment status), as well as environmental circumstances and healthcare system resources [8,9,10]. In this context, studies have consistently shown a strong association between maternal socioeconomic status and personal characteristics with both the initiation and continuation of breastfeeding. Women with lower education levels and unstable employment are significantly less likely to start or continue breastfeeding [10,11,12]. Moreover, contextual and regional inequalities have a substantial impact on breastfeeding outcomes. Women living in disadvantaged areas were more likely to interrupt exclusive breastfeeding, indicating that environmental and structural barriers may outweigh individual characteristics in shaping breastfeeding practices [13]. Although still largely underexplored, recent research has begun to shed light on the proactive role of fathers in supporting breastfeeding outcomes, including improved initiation, duration, and exclusivity [14,15,16]. Paternal engagement may provide meaningful behavioral support to lactating women and contribute to early childcare, thereby enhancing maternal confidence in breastfeeding and positively influencing maternal mental health and self-efficacy [17]. For all these reasons, understanding the interplay between parental sociodemographic factors and breastfeeding behavior is also essential for addressing inequalities and informing public health strategies that promote equitable adherence to best practices.

This study provides novel contributions to the literature by examining breastfeeding determinants in a large cohort of mother-child pairs from Eastern Sicily enrolled in the MAMI-MED study. The cohort includes longitudinal follow-up data covering breastfeeding initiation, exclusivity, and duration, allowing a more detailed temporal characterization than most previous Italian studies. Furthermore, our analysis incorporates both maternal and paternal sociodemographic characteristics, an aspect rarely investigated in the Italian context, thereby offering a more comprehensive view of the familial and social determinants of breastfeeding practices.

## 2. Materials and Methods

### 2.1. The MAMI-MED Cohort

The present analysis was conducted within the “MAMI-MED” cohort, a prospective study initiated in December 2020 with the objective of evaluating the influence of social, environmental, behavioral, and molecular factors on the health of mother–child dyads. The study protocol aligns with the methodology of the “Mamma and Bambino” cohort, which has been ongoing in Catania since 2015 [12,18,19,20]. Participants were recruited among pregnant women attending their first-trimester prenatal visit as part of routine care at the Azienda di Rilievo Nazionale e di Alta Specializzazione (ARNAS) Garibaldi Nesima Hospital in Catania, Italy. Mother–child pairs with pre-existing medical conditions (i.e., autoimmune and/or chronic diseases), pregnancy complications (i.e., preeclampsia, hypertension, and diabetes), pre-term induced delivery or cesarean section, intrauterine fetal death, plurality, and congenital malformations were excluded. The MAMI-MED study protocol received approval from the Catania 2 Ethics Committee (Protocol Numbers: 487/CE, 71/2020/CECT2, 157/CEL) and was conducted in accordance with the principles of the Declaration of Helsinki. All participants were thoroughly informed about the purpose and procedures of the study and provided written informed consent prior to their inclusion.

### 2.2. Study Population and Data Collection

Between December 2020 and October 2023, 1350 pregnant women attending their first-trimester prenatal visit at the ARNAS Garibaldi Nesima Hospital in Catania were screened for eligibility. After applying the exclusion criteria, 1318 were enrolled in the MAMI-MED cohort. Of these, 1312 completed the one-year postpartum follow-up, and 829 also participated in the two-year follow-up; 5 cases were excluded from specific analyses due to missing or inconsistent breastfeeding data. Information on maternal and paternal health status, socioeconomic and personal characteristics, behavioral factors, and child health outcomes and related habits was collected by trained epidemiologists at baseline (i.e., during pregnancy) and during scheduled telephone interviews at delivery, and at 12 and 24 months postpartum, using a standardized and structured questionnaire. Parental educational level was categorized as low (primary education; ≤8 years of schooling), medium (secondary education; ≤13 years), and high (more than 13 years of schooling). Maternal and paternal employment status was classified as employed (both full-time and part-time) or unemployed, the latter category including students and housewives. Regarding smoking habits, women were categorized as non-smokers or smokers, with the latter including both current and former smokers. At recruitment, women were also asked to self-report their pre-pregnancy weight and height, which were used to calculate the pre-pregnancy body mass index (BMI), defined as weight in kilograms divided by height in meters squared. Pre-gestational BMI was classified according to World Health Organization (WHO) criteria as underweight (<18.5), normal weight (18.5–24.9), overweight (25–29.9), and obese (≥30) [21]. Gestational weight gain (GWG) was classified as inadequate, adequate, or excessive according to pre-pregnancy BMI categories, following the 2009 Institute of Medicine (IOM) guidelines, which recommend total weight gains of 12.5–18 kg for underweight women, 11.5–16 kg for those with normal weight, 7–11.5 kg for overweight women, and 5–9 kg for obese women [22]. In addition, data on obstetric history and delivery outcomes were collected, including parity (nulliparous vs. multiparous), type of delivery (vaginal vs. cesarean), and gestational age at recruitment.

### 2.3. Breastfeeding Data

Information on breastfeeding practices was collected through an ad hoc questionnaire administered via telephone interviews at the 12- and 24-month follow-ups after birth. At the 12-month follow-up, women were asked whether they had breastfed, with a “yes” indicating that they had breastfed or at least attempted to do so, and a “no” indicating that they had never breastfed nor attempted to breastfeed. We also collected information on the duration of breastfeeding, whether formula feeding had been introduced, and, if applicable, the age at which it was initiated (in months). Based on these data, we derived each woman’s breastfeeding regimen, classifying it as exclusive breastfeeding during the first six months (infant receives only breast milk and no other liquids or solids, except for oral rehydration solution, drops, or syrups of vitamins, minerals, or medicines), predominant breastfeeding (breast milk is the main source of nourishment, but the infant also receives water or water-based drinks, juices, or ritual fluids—small amounts of liquids given for cultural or religious purposes—and no non-human milk or food-based fluids), or mixed feeding (breast milk is given along with non-human milk, formula, or solid/semi-solid foods). Accordingly, women were considered adherent to the WHO recommendations on breastfeeding if they exclusively breastfed for the first six months of life (i.e., without introducing any other foods or liquids) [23]. Furthermore, women were asked whether they were still breastfeeding. During the 24-month follow-up interview, women were further asked whether they had continued breastfeeding up to 24 months or beyond.

### 2.4. Statistical Analyses

All statistical analyses were conducted using SPSS software, version 26.0 (SPSS Inc., Chicago, IL, USA). The assumption of normality for continuous variables was assessed via the Kolmogorov–Smirnov test. Descriptive statistics were reported as medians and interquartile ranges (IQRs) for continuous variables, and as absolute and relative frequencies for categorical variables. Non-normally distributed continuous variables were compared between groups using the Mann–Whitney U test. Associations between categorical variables were evaluated using Pearson’s Chi-square test. Multivariable logistic regression analyses were performed to estimate the independent associations between covariates and two binary outcomes: (1) breastfeeding status, and (2) adherence to the WHO recommendation of exclusive breastfeeding for six months among women who initiated breastfeeding. Variables identified as significant at the bivariate level were included in the multivariable models. All models were adjusted for maternal age to account for potential confounding. Regression coefficients were reported as odds ratios (ORs) with 95% confidence intervals (CIs). All hypothesis tests were two-tailed, and statistical significance was defined as *p* < 0.05.

## 3. Results

### 3.1. Characteristics of the MAMI-MED Cohort

Table 1 presents the characteristics of the 1312 women in the MAMI-MED cohort (median age = 31 years; IQR = 6) who completed follow-up interviews at childbirth and at the first year of follow-up. At the time of recruitment, 45.7% of the women had at least one child in addition to the one involved in the present study. Regarding maternal socioeconomic characteristics, 75.4% of participants reported a medium or high education level, and 51.0% were employed either full-time or part-time. Based on pre-pregnancy BMI (mean = 23.4; SD = 6.04), 58.4% of the women were classified as normal weight. Adequate GWG was reported by 32.8% of the cohort. With respect to maternal lifestyle factors, 9.5% of the women were current smokers. As for partner characteristics, 34.7% had a low educational level, 50.3% medium, and 15.0% high; the majority (93.3%) were employed. Among the 1312 women included in the analysis, 78.3% (n = 1027) reported having initiated breastfeeding, while the remaining 21.7% (n = 285) did not breastfeed at all, including no reported attempts to initiate breastfeeding. Among those who breastfed, 29.6% (n = 304) exclusively breastfed during the first six months, 69.7% (n = 716) practiced mixed feeding, and 0.7% (n = 7) reported predominant breastfeeding. Within this sample, 829 women also completed the follow-up interview at two years postpartum. In this subgroup, 143 (17.2%) were still breastfeeding on the child’s first birthday, and notably, 60 of them (41.9%) continued breastfeeding up to 24 months or beyond.

### 3.2. Parental Characteristics Associated to Breastfeeding

Table 1 summarizes maternal and partner characteristics according to breastfeeding status. Compared to mothers who did not breastfeed, those who breastfed were slightly older (median age: 31 vs. 30 years; *p* = 0.016), and more likely to hold a higher educational level (29.1% vs. 10.9%; *p* < 0.001) and to be employed (53.6% vs. 41.9%; *p* < 0.001). Breastfeeding mothers also exhibited a lower mean pre-gestational BMI (23.2 vs. 24.4; *p* < 0.001) and a lower prevalence of obesity (11.1% vs. 20.4%). Additionally, they were less likely to be current smokers (8.5% vs. 13.0%; *p* = 0.022) and more likely to have had a vaginal delivery (72.9% vs. 61.8%; *p* < 0.001). Interestingly, we noted that partner-related factors were also associated with breastfeeding. Women who breastfed were more likely to have a partner with a medium or high level of education (68.9% vs. 52.3%; *p* < 0.001) and an employed partner (94.0% vs. 90.5%; *p* = 0.036).

In line with these findings, our logistic regression model showed (Table 2) that women with a medium (OR = 1.781; 95% CI: 1.258–2.521; *p* = 0.001) and a high level of education (OR = 3.892; 95% CI: 2.255–6.718; *p* < 0.001), as well as those who had a spontaneous delivery (OR = 1.461; 95% CI: 1.090–1.958; *p* = 0.011), were more likely to breastfeed compared to their less educated counterparts and those who underwent a cesarean section. These results were obtained by adjusting for maternal age, employment status, smoking status, pre-gestational BMI, nutritional status, partner educational level, and partner occupational status.

### 3.3. Characteristics of Women Adhering to WHO Breastfeeding Recommendations

Among the women who reported initiating breastfeeding, 304 (29.6%) adhered to the WHO recommendation of exclusive breastfeeding for the first six months. Compared to non-adherent mothers (Table 3), adherent women were significantly more likely to have a high level of education (38.1% vs. 25.3%; *p* < 0.001) and to be employed (58.9% vs. 51.4%; *p* = 0.027). Furthermore, the prevalence of current smoking was significantly lower among adherent women (3.9% vs. 10.4%; *p* < 0.001). Although no significant differences were found in mean pre-gestational BMI or nutritional status, adherent mothers were significantly more likely to report adequate gestational weight gain (38.4% vs. 31.7%; *p* = 0.012). No statistically significant differences emerged in maternal age, parity, or mode of delivery. Additionally, women in the adherent group more frequently had partners with a higher level of education (24.8% vs. 13.8%; *p* < 0.001) and stable employment (96.7% vs. 92.9%; *p* = 0.020).

In the logistic regression model (Table 4), we found that women with a medium education (OR = 2.144; 95% CI: 1.339–3.433; *p* = 0.002), and those with a high education level had a higher odd of being adherent than their counterparts (OR = 2.611; 95% CI: 1.489–4.580; *p* < 0.001). Furthermore, non-smoking women were significantly more likely to be adherent compared to smokers (OR = 2.256; 95% CI: 1.158–4.395; *p* = 0.017). Regarding GWG, women with adequate GWG showed higher odds of adherence (OR = 1.506; 95% CI: 1.035–2.189; *p* = 0.032), whereas the association with reduced GWG did not reach statistical significance. Although not statistically significant, the employment status of the father showed a borderline association with adherence. Specifically, children of employed fathers were more than twice as likely to be adherent compared to those of unemployed fathers (OR = 2.124; 95% CI: 0.968–4.658; *p*= 0.060). Other variables included in the model (i.e., maternal age, parental employment status, and paternal education) were not significantly associated with adherence.

## 4. Discussions

Breast milk provides all the energy and nutrients to meet an infant’s nutritional needs, also providing maternal health benefits [24]. As recommended by WHO, exclusive breastfeeding during the first six months of life is a key preventive strategy against obesity and non-communicable diseases later in life [4,25]. Evidence suggests that early cessation of breastfeeding is associated with excessive weight gain in infancy, while exclusive breastfeeding is associated with healthier long-term outcomes [26,27]. In addition to its nutritional and developmental benefits, human breast milk has the potential to influence epigenetic mechanisms and the expression of genes involved in infant metabolic pathways and immune functions [28]. However, at a global level, the ambitious goal of achieving 70% exclusive breastfeeding in the first six months by 2030 remains far to be reached [29]. This scenario highlights the urgent need for specific public strategies that address the diverse sociocultural, economic, and structural barriers influencing maternal adherence to breastfeeding practices.

In our cohort, 78.3% of women reported having breastfed, but only 29.6% practiced exclusive breastfeeding during the first six months. These figures are consistent with national surveillance data indicating lower exclusive breastfeeding rates in southern Italy compared to the north [7]. Such findings suggest that, although national legislation provides important structural support through maternity leave and breastfeeding protections, legal measures alone may be insufficient to ensure high adherence to breastfeeding recommendations. The patterns observed in our population likely reflect the combined influence of individual sociodemographic characteristics and broader structural factors, including limited healthcare resources and persistent socioeconomic disparities in the region. Our results therefore underscore the need to complement national policies with targeted, context-specific interventions that address environmental and social barriers to breastfeeding continuation, particularly in disadvantaged areas.

Notably, compared with non-breastfeeding mothers, those who initiated breastfeeding displayed a more favorable sociodemographic and behavioral profile. They were slightly older, more educated, more likely to be employed, had lower pre-pregnancy BMI and obesity rates, were less likely to smoke, and more frequently delivered vaginally. These associations reinforce the role of medical, behavioral, and socioeconomic factors in shaping breastfeeding experiences and outcomes. These results were also confirmed by logistic regression, in which women with a medium and a high level of education, as well as those who had a spontaneous delivery, were more likely to breastfeed compared to their less educated counterparts and those who underwent a cesarean section. These results may reflect the broader advantages associated with older age and higher educational attainment, which often translate into greater health literacy, more proactive engagement with prenatal care, and stronger adherence to public health recommendations. In particular, higher maternal education is a well-established marker of elevated socioeconomic status, where access to healthcare services is not only more frequent but also more effectively utilized [30]. In line, a recent meta-analysis confirmed that maternal awareness of breastfeeding benefits, higher educational attainment, and vaginal delivery were positively associated with exclusive breastfeeding [31,32]. Our findings reinforce existing evidence that higher age, education, and social support, are key drivers of breastfeeding motivation.

Both intrinsic factors (e.g., the desire to provide the best for the infant) and extrinsic influences (e.g., support from family and healthcare professionals) play a critical role [33]. Studies also suggested that sociodemographic determinants, maternal lifestyle factors—such as smoking status, physical activity, and overall health behaviors—also significantly influence breastfeeding initiation and duration [34,35]. Additionally, vaginal delivery facilitates early mother-infant bonding and timely initiation of breastfeeding, which are critical for establishing lactation [34]. In our cohort, women adherent to exclusive breastfeeding for the first six months were also significantly more likely to have a high level of education and to be employed. This aligns with recent European findings showing that higher maternal education and formal employment are associated with exclusive breastfeeding continuation after returning to work [36]. Studies suggested that women with a higher educational level were more likely to follow preventive behaviors, including prolonged breastfeeding duration, attentive child care, and active use of health services [37]. We also found that adherent mothers showed a significantly lower prevalence of smoking. This is consistent with broad literature demonstrating that smoking reduces prolactin levels and milk supply, thereby decreasing both the probability and the duration of exclusive breastfeeding [38]. Moreover, a previous analysis within the MAMI-MED cohort identified two different profiles based on education level, employment status, pre-pregnancy nutritional status, and Mediterranean Diet Score (MDS). Cluster 1 comprised women with lower educational attainment, who were unemployed, overweight and/or obese, and had a lower mean MDS. Cluster 2 was predominantly composed of women with medium-to-high educational levels, who were employed, had normal weight, and reported a higher mean MDS. Women in Cluster 1 showed significantly higher proportions of preterm births, low-birth-weight newborns, and large-for-gestational-age newborns, as well as lower mean gestational age, birth weight, and newborn length compared to Cluster 2. These findings suggest that socioeconomic advantage and healthier lifestyle behaviors tend to coexist and are associated with more favorable pregnancy and neonatal outcomes, whereas socioeconomic disadvantage is linked to less healthy lifestyles and poorer perinatal health indicators [6].

In the current analysis, while no significant differences were found in pre-pregnancy BMI, women who adhered to exclusive breastfeeding recommendations were significantly more likely to report adequate gestational weight gain, which may reflect intentional, conscious behaviors that facilitate sustained breastfeeding [39]. Also, the logistic regression model suggested that medium and high education levels, non-smokers and having adequate GWG were the main predictors of the WHO breastfeeding recommendations. Interestingly, our study also highlights the crucial role of the partner, particularly their socioeconomic characteristics in influencing both the decision to initiate breastfeeding and adherence to exclusive breastfeeding during the first six months. Women who breastfed, as well as those who adhered to exclusive breastfeeding recommendations, were more likely to have a partner with a higher level of education and stable employment. These findings suggest that a socioeconomically stable partner contributes to a supportive and sociocultural enabling family environment, which facilitates breastfeeding initiation and continuation. Our findings align with the WHO’s perspective, which emphasizes that breastfeeding becomes easier when the partner is actively involved and shares the experience [40].

Our study has several limitations that should be considered. First, breastfeeding data were self-reported, which may have introduced recall or reporting bias. Discrepancies between responses collected at the 1-year and 2-year follow-ups led to the exclusion of inconsistent cases, resulting in a reduced sample size. Second, the low proportion of children breastfed up to 24 months limited our ability to investigate factors potentially associated with prolonged breastfeeding. Third, no direct measures of maternal or household socioeconomic status were collected, despite its well-documented influence on breastfeeding practices. In Italy, the collection of detailed socioeconomic information (e.g., household income) is often restricted by strict privacy regulations and low participant acceptability, reducing its feasibility in observational studies. Furthermore, changes in maternal or paternal socioeconomic conditions over time were not assessed, as variables such as return to work, employment type, or financial status were only collected at recruitment and not updated during follow-up. Fourth, other potentially relevant behavioral factors, such as maternal diet and physical activity, were not included in the analysis. In addition, the exclusively quantitative design precluded the integration of qualitative elements—such as perceived barriers, attitudes, and cultural beliefs—which could have enriched the understanding of the factors influencing breastfeeding. Incorporating these dimensions in future research would allow for a more comprehensive assessment of the phenomenon. Finally, the definition of our primary outcome may have introduced classification bias. The breastfeeding group included all mothers who reported initiating breastfeeding, even if for only a very brief period (e.g., less than one month). Although data on breastfeeding duration were collected, missing information for this variable in several cases limited the possibility of performing more detailed analyses based on breastfeeding length.

## 5. Conclusions

These findings, together with the observed associations between maternal and paternal characteristics and breastfeeding practices, highlight the importance of creating supportive environments that enable families to make informed and sustained choices. Within this framework, educational attainment, health-motivated behaviors, and stable socioeconomic conditions emerge as key determinants of adherence to breastfeeding recommendations. Targeted interventions addressing specific sociodemographic groups could therefore yield substantial public health benefits. Moreover, exploring possible underlying factors—such as changes in maternity leave policies, increasing medicalization of childbirth, evolving cultural norms, or a decline in institutional support for breastfeeding women—could help clarify the structural determinants of suboptimal breastfeeding rates and inform the design of more effective, context-specific strategies.

## Figures and Tables

**Table 1 nutrients-17-02702-t001:** Association between maternal characteristics and breastfeeding practices.

Characteristics	Overall (n = 1312)	Breastfeeding (n = 1027)	Not Breastfeeding (n = 285)	*p*-Value
Age (Years)	31.0 (6.0)	31.0 (6.0)	30.0 (6.0)	0.016
Educational Level				
Low	24.5%	20.3%	39.3%	<0.001
Medium	50.3%	50.6%	49.3%	
High	25.1%	29.1%	10.9%	
Employment				
Yes	51.1%	53.6%	41.9%	<0.001
Not	48.9%	46.4%	58.1%	
Smoker				
Yes	9.5%	8.5%	13.0%	0.022
Not	90.5%	91.5%	87.0%	
Pre-Gestational BMI	23.4 (6.04)	23.2 (5.7)	24.4 (7.4)	<0.001
Nutritional Status				
Underweight	5.4%	5.7%	4.2%	<0.001
Normal Weight	58.4%	60.9%	49.5%	
Overweight	23.1%	22.3%	26.0%	
Obese	13.1%	11.1%	20.4%	
Gestational Weight Gain ^a^				
Reduced	38.8%	39.3%	36.7%	0.078
Adequate	32.8%	33.7%	29.5%	
Excessive	28.5%	27.0%	33.8%	
Having Children ^b^				
Yes	45.7%	45.5%	46.5%	0.780
No	54.3%	54.5%	53.5%	
Type of Delivery				
Natural	70.6%	72.90%	61.8%	<0.001
Cesarean	29.4%	27.1%	38.2%	
Partner Educational Level				
Low	34.7%	31.1%	47.7%	<0.001
Medium	50.3%	51.8%	44.9%	
High	15.0%	17.1%	7.4%	
Partner Occupational Status				
Employed	93.3%	94.0%	90.5%	0.036
Unemployed	6.7%	6.0%	9.5%	

Data are presented as median (interquartile range) or frequency (percentage). Comparisons were performed using the Mann–Whitney U test or Chi-squared test, as appropriate. ^a^ Categorized based on pre-pregnancy BMI and the Institute of Medicine guidelines. ^b^ Defined as having at least one other child besides the one included in this study.

**Table 2 nutrients-17-02702-t002:** Logistic regression of the association between maternal characteristics and breastfeeding status.

Characteristics	OR ^a^	95% CI	*p*-Value
Educational Level			
Low	Ref.		
Medium	1.781	1.258–2.521	*p* = 0.001
High	3.892	2.255–6.718	*p* < 0.001
Type of Delivery			
Cesarean	Ref.		
Natural	1.461	1.090–1.958	*p* = 0.011

^a^ Results are expressed as odds ratios (OR) with 95% confidence intervals (CI) and corresponding *p*-values. “Ref” indicates the reference category used for each categorical variable in the model. The dependent variable in the logistic regression model was breastfeeding status, defined as whether or not the mother breastfed. Independent variables included maternal age (in years), educational level (low, medium, high), employment status (employed/unemployed), smoking status during pregnancy (yes/no), pre-gestational body mass index (BMI), nutritional status (underweight, normal weight, overweight, obese), mode of delivery (natural or cesarean), partner educational level (low, medium, high), and partner occupational status (employed or unemployed).

**Table 3 nutrients-17-02702-t003:** Association between parental characteristics and compliance with breastfeeding recommendations.

Characteristics	Non-Adherent (n= 723)	Adherent (n = 304)	*p*-Value
Age (Years)	31.0 (6.0)	31.0 (5.0)	0.058
Educational Level			
Low	24.2%	10.9%	<0.001
Medium	50.5%	51.0%	
High	25.3%	38.1%	
Employment			
Yes	51.4%	58.9%	0.027
Not	48.6%	41.1%	
Current Smoker			
Yes	10.4%	3.9%	<0.001
Not	89.6%	96.1%	
Pre-Gestational BMI	23.4 (5.9)	22.9 (5.1)	0.156
Nutritional Status			
Underweight	5.8%	5.3%	0.078
Normal Weight	58.8%	66.0%	
Overweight	22.8%	21.0%	
Obese	12.6%	7.7%	
Gestational Weight Gain ^a^			
Reduced	38.8%	40.7%	0.012
Adequate	31.7%	38.4%	
Excessive	29.6%	20.9%	
Having Children ^b^			
Yes	44.7%	47.5%	0.409
No	55.3%	52.5%	
Type of Delivery			
Natural	72.1%	75.3%	0.281
Cesarean	27.9%	24.7%	
Partner Educational Level			
Low	33.9%	24.5%	<0.001
Medium	52.3%	50.7%	
High	13.8%	24.8%	
Partner Occupational Status			
Employed	92.9%	96.7%	0.020
Unemployed	7.1%	3.3%	

Data are presented as median (interquartile range) or frequency (percentage). Comparisons were performed using the Mann–Whitney U test or Chi-squared test, as appropriate. ^a^ Categorized based on pre-pregnancy BMI and the Institute of Medicine guidelines. ^b^ Defined as having at least one other child besides the one included in this study.

**Table 4 nutrients-17-02702-t004:** Logistic regression of the association between maternal characteristics and compliance with breastfeeding recommendations.

Characteristics	OR ^a^	95% CI	*p*-Value
Educational Level			
Low	Ref.		
Medium	2.144	1.339–3.433	*p* = 0.002
High	2.611	1.489–4.580	*p* < 0.001
Current Smoker			
Yes	Ref		
Not	2.256	1.158–4.395	*p* = 0.017
Gestational Weight Gain			
Excessive	Ref		
Reduced	1.345	0.931–1.944	*p* = 0.115
Adequate	1.506	1.035–2.189	*p* = 0.032

^a^ Results are expressed as odds ratios (OR) with 95% confidence intervals (CI) and corresponding *p*-values. “Ref” indicates the reference category used for each categorical variable in the model. The dependent variable in the logistic regression model was adherence to WHO breastfeeding recommendations (yes/no), defined as whether or not the mother breastfed. Independent variables included maternal age (in years), educational level (low, medium, high), employment status (employed/unemployed), smoking status during pregnancy (yes/no), gestational weight gain (reduced, adequate, excessive), partner educational level (low, medium, high) and partner occupational status (employed or unemployed).

## Data Availability

The data presented in this study are available on request from the corresponding due to privacy and ethical restrictions.

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
