# Peer review of "Breastfeeding and Sociodemographic Determinants: Evidence from the “MAMI-MED” Cohort"

_nutrients, 2025, doi:10.3390/nu17162702_

Round 1
Reviewer 1 Report
Comments and Suggestions for Authors
Dear Authors,
The following remarks and comments are intended to strengthen the manuscript and make it more understandable to the reader:
Abstract:
Information about the time period in which the studies were conducted and the number of women involved must be provided.
Introduction:
In this section, it is advisable to provide information regarding the legal regulations concerning the promotion of breastfeeding, such as the length of maternity leave, breaks for feeding, or reduced working hours, etc.
In my opinion, it is important to emphasise how the conducted study contributes new content to the area of knowledge about the determinants of breastfeeding. For example, what distinguishes this study from others (e.g. sample size, duration of observation, etc.). In addition, I propose the introduction of paragraph breaks to enhance the text's navigability.
Materials and methods:
It would be beneficial to include the research diagram, taking into account the time and number of participants.
Lines 126-128: For the convenience of the reader, IOM recommendations should be provided, depending on pre-pregnancy BMI.
The criteria for including women in the cohort are not clearly defined. I am aware that this information was published previously, however, for the reader's convenience, it would be beneficial to include this information either in the text of the manuscript or in the supplementary materials (e.g. questionnaire).
Authors should provide an explanation of the meaning of 'predominant breastfeeding' in comparison to 'mixed feeding'.
Results:
It is not entirely clear to me whether the women described as 'not breastfeeding' in Table 1 (n=285) are women who have never breastfed or made any attempt to breastfeed. If that is the case, then that is quite a significant percentage. Please clarify.
In Table 2, it would be preferable to provide the references first in the case of the type of birth (in a similar manner to the way in which the education level is presented).
I am unable to discern the distinction between 'compliance with breastfeeding recommendations' in Table 3 and 'adherence to WHO breastfeeding recommendations' (Table 4). Could you please clarify this point or use the same expression?
General remark: It is believed that the socio-economic status of a mother (family) is a significant factor influencing breastfeeding practices. However, the authors did not consider this in their analyses. What is the reason for this?
Discussion
I have significant reservations about the authors' ability to distinguish women with a 'healthy lifestyle' (e.g. lines 350-351) based on the data presented, as they have not analysed key factors such as diet (including alcohol consumption), physical activity, or sleep duration. In my professional opinion, it is challenging to assess the healthiness of a lifestyle solely based on information about body weight and smoking.
Please note: There is some difficulty in reading the uniform text. It is advisable for authors to consider dividing it into paragraphs.
Conclusions:
In this section, the authors mention the shaping of early nutritional behaviors. However, it seems that they should limit their conclusions solely to breastfeeding, as they did not assess how infants are fed, including, for example, the type and timing of introduced foods.
Author Response
Dear Editor, we thank both reviewers for their constructive comments and valuable suggestions, which have helped us to improve the clarity, completeness, and interpretative strength of our manuscript. Below we provide a point-by-point response to each comment. All changes made to the manuscript are highlighted in red in the revised version.
Reviewer 1
Comment: Information about the time period in which the studies were conducted and the number of women involved must be provided.
Response: We have added the time period of data collection and the number of participants to the Abstract.
Comment: In this section, it is advisable to provide information regarding the legal regulations concerning the promotion of breastfeeding, such as the length of maternity leave, breaks for feeding, or reduced working hours, etc.
Response: We have added a brief description of the Italian legal framework for breastfeeding promotion, including maternity leave, breastfeeding breaks, and protections for working mothers.
Comment: It is important to emphasise how the conducted study contributes new content to the area of knowledge about the determinants of breastfeeding. For example, what distinguishes this study from others (e.g. sample size, duration of observation, etc.). In addition, I propose the introduction of paragraph breaks to enhance the text's navigability.
Response: We have clarified the novelty of our study, highlighting the large, region-specific cohort, the inclusion of paternal characteristics, and the longitudinal follow-up. We have also introduced paragraph breaks throughout the Introduction for better readability.
Comment: It would be beneficial to include the research diagram, taking into account the time and number of participants.
Response: Instead of a diagram, we have added a detailed description of the participant flow, including time frame, number assessed for eligibility, exclusions, and follow-up completion.
Comment: Lines 126–128: For the convenience of the reader, IOM recommendations should be provided, depending on pre-pregnancy BMI.
Response: We have now specified the IOM 2009 recommendations for GWG by pre-pregnancy BMI category.
Comment: The criteria for including women in the cohort are not clearly defined. I am aware that this information was published previously, however, for the reader's convenience, it would be beneficial to include this information either in the text of the manuscript or in the supplementary materials (e.g. questionnaire).
Response: We have now detailed the exclusion criteria within the Materials and Methods section.
Comment: Authors should provide an explanation of the meaning of 'predominant breastfeeding' in comparison to 'mixed feeding'.
Response: We have added definitions of exclusive, predominant, and mixed breastfeeding to the Methods.
Comment: It is not entirely clear whether the women described as 'not breastfeeding' are those who have never breastfed or made any attempt to breastfeed.
Response: We have clarified that these women never initiated breastfeeding.
Comment: In Table 2, it would be preferable to provide the references first in the case of the type of birth.
Response: Table 2 has been reformatted to present reference categories first for the type of birth.
Comment: Clarify the distinction between 'compliance with breastfeeding recommendations' (Table 3) and 'adherence to WHO breastfeeding recommendations' (Table 4).
Response: We have harmonized terminology across the manuscript to avoid confusion.
Comment: It is believed that the socio-economic status of a mother (family) is a significant factor influencing breastfeeding practices. However, the authors did not consider this in their analyses. What is the reason for this?
Response: We have explained in the Limitations that detailed socio-economic data were not collected due to privacy constraints and low participant acceptability in the Italian context.
Comment: I have significant reservations about the authors' ability to distinguish women with a 'healthy lifestyle' (e.g. lines 350-351) based on the data presented, as they have not analysed key factors such as diet (including alcohol consumption), physical activity, or sleep duration. In my professional opinion, it is challenging to assess the healthiness of a lifestyle solely based on information about body weight and smoking.
We agree and have revised the text to avoid overinterpretation, specifying the lifestyle variables available in our analyses.
Comment: There is some difficulty in reading the uniform text. It is advisable for authors to consider dividing it into paragraphs
Response: We agree that paragraph breaks could improve readability. However, the journal’s formatting guidelines do not allow the Discussion to be divided into subsections. We have nevertheless revised the text to improve its navigability by restructuring sentences, improving transitions, and shortening long passages where appropriate.
Comment: In this section, the authors mention the shaping of early nutritional behaviors. However, it seems that they should limit their conclusions solely to breastfeeding, as they did not assess how infants are fed, including, for example, the type and timing of introduced foods.
Response: Conclusions have been revised accordingly.
Reviewer 2 Report
Comments and Suggestions for Authors
This article analyses maternal and paternal determinants associated with the practice and adherence to WHO recommendations on breastfeeding, with data from the “MAMI-MED” cohort. The study reinforces already established evidence and adds value by emphasizing the positive role of the partner in breastfeeding and demonstrating the importance of maternal and paternal socioeconomic profiles.
The study methodology is adequate and consistent, although the data on the main variable “exclusive breastfeeding” are self-reported and have a risk of memory bias, potentially overestimating the real prevalence.
The analysis is exclusively quantitative and the lack of qualitative elements, such as: barriers, attitudes, cultural beliefs would further enrich the understanding of the phenomenon.
In the discussion, the lack of national contextualization: Although the article mentions regional differences, it does not discuss the reasons for the reduction in breastfeeding rates in Italy in recent years. This decline is documented in national surveillance sources, and its omission compromises the critical contextualization of the findings. It would be relevant to explore possible explanations — such as changes in maternity leave policies, increasing medicalization of childbirth, cultural changes or a decline in institutional support for breastfeeding women — that could help understand the structural determinants of this negative trend.

Author Response
Dear Editor, we thank both reviewers for their constructive comments and valuable suggestions, which have helped us to improve the clarity, completeness, and interpretative strength of our manuscript. Below we provide a point-by-point response to each comment. All changes made to the manuscript are highlighted in red in the revised version.
Reviewer 2
Comment: The study methodology is adequate and consistent, although the data on the main variable “exclusive breastfeeding” are self-reported and have a risk of memory bias, potentially overestimating the real prevalence.
Response: We acknowledge this limitation and have included it in the Limitations section.
Comment: The analysis is exclusively quantitative; the lack of qualitative elements limits the understanding of the phenomenon.
Response: We agree and have acknowledged this as a limitation, suggesting future studies should include qualitative data.
Comment: In the discussion, the lack of national contextualization: Although the article mentions regional differences, it does not discuss the reasons for the reduction in breastfeeding rates in Italy in recent years. This decline is documented in national surveillance sources, and its omission compromises the critical contextualization of the findings. It would be relevant to explore possible explanations — such as changes in maternity leave policies, increasing medicalization of childbirth, cultural changes or a decline in institutional support for breastfeeding women — that could help understand the structural determinants of this negative trend.
Response: We have added discussion on national trends, referencing documented declines in exclusive breastfeeding rates, and considered possible explanations, including changes in maternity leave policies, increased medicalization of childbirth, evolving cultural norms, and reduced institutional support.
Round 2
Reviewer 1 Report
Comments and Suggestions for Authors
Thanks to the Authors for replying my comments.
I would only suggest to explain the meaning of " ritual fluids" (line 172) as it is not clear.
Author Response
As requested, we have defined the meaning of ritual liquids.